# The Land surface Data Toolkit (LDT v7.2) – a data fusion environment for land data assimilation systems

Kristi R. Arsenault [1,2], Sujay V. Kumar [2], James V. Geiger [3], Shugong Wang[1,2], Eric Kemp[4,2], David M. Mocko [1,2], Hiroko Kato Beaudoing [5,2], Augusto Getirana [5,2], Mahdi Navari [5,2], Bailing Li [5,2], Jossy Jacob [4,2], Jerry Wegiel [1,6] and Christa D. Peters-Lidard [7]

[1] Science Applications International Corporation, McLean, VA, USA

[2] Hydrological Sciences Laboratory, NASA Goddard Space Flight Center, Greenbelt, MD, USA

[3] Science Data Processing Branch, NASA Goddard Space Flight Center, Greenbelt, MD, USA

[4] Science Systems and Applications, Inc., Lanham, MD, USA

[5] ESSIC, University of Maryland, College Park, MD, USA

[6] Headquarters 557th Weather Wing, Offutt Air Force Base, NE, USA

[7] Earth Sciences Division, NASA Goddard Space Flight Center, Greenbelt, MD USA

*Correspondence to:* K. R. Arsenault (Kristi.R.Arsenault@nasa.gov)

**Abstract.** The effective applications of land surface model (LSM) and hydrologic models pose a varied set of data input and processing needs, ranging from ensuring consistency checks to more derived data processing and analytics. This article describes the development of the Land surface Data Toolkit (LDT), which is an integrated framework designed specifically for processing input data to execute LSMs and hydrological models. LDT not only serves as a pre-processor to the NASA Land Information System (LIS), which is an integrated framework designed for multi-model LSM simulations and data assimilation (DA) integrations, but also as a land surface-based observation and DA input processor. It offers a variety of user options and inputs to processing datasets for use within LIS and stand-alone models. The LDT design facilitates the use of common data formats and conventions. LDT is also capable of processing LSM initial conditions, meteorological boundary conditions and ensuring data quality for inputs to LSMs and DA routines. The machine learning layer in LDT facilitates the use of modern data science algorithms for developing data-driven predictive models. Through the use of an object-oriented framework design, LDT provides extensible features for the continued development of support for different types of observational data sets and data analytics algorithms to aid land surface modelling and data assimilation.

## 1. Introduction

Accurate quantification of terrestrial water and energy cycles is important for a wide range of applications including weather and climate modelling and initialization, agricultural and water management, estimation of hydrological hazards such as droughts and floods, among others. The need for robust estimates of land surface conditions to support these applications have led to the development of Land Data Assimilation Systems (LDASs; e.g., Rodell et al., 2004; Mitchell et al., 2004; Chen et al., 2007). The key emphasis of an LDAS is the integration of the state-of-the art land surface models (LSMs) with high quality observations from in situ networks, reanalyses and remote sensing, in order to obtain an improved representation of land surface processes. The synthesis of several types of model and observation data across various spatial and temporal resolutions and extents is needed to support the development of flexible LDAS configurations for conducting both research and application-oriented studies.

The LDAS environments pose a varied set of data synthesis requirements based on the modelling configurations. The models within LDASs are typically executed in an uncoupled fashion by isolating the land surface and by providing the required boundary conditions from the atmosphere. These conditions are derived from the outputs of atmospheric models, remote sensing and ground observations. The LSMs also require specifications of land surface characteristics such as vegetation, soils and topography, which are a mix of both time-invariant and time-varying parameters. The data assimilation (DA) tools in LDASs incorporate the information from remote-sensing and ground observations to constrain and improve model states. Similarly, the optimization and uncertainty estimation tools exploit observational information to calibrate and estimate the uncertainty associated with the model parameters. In addition to these external data needs, data processing requirements related to initialization, spatial and temporal disaggregation and bias mitigation are also often encountered in LDAS modelling scenarios. Finally, there is often a significant technology gap in bridging the technical advances in data science and processing methods with the land modelling approaches.

These challenges and gaps have motivated the development of a data fusion environment known as the Land surface Data Toolkit (LDT). The primary function of LDT is to serve as a data synthesis environment for terrestrial LDASs. LDT is currently designed as the preprocessor to the NASA Land Information System (LIS; Kumar et al. 2006, Peters-Lidard et al. 2007), which is an open source software infrastructure for land surface modelling and designed to facilitate the efficient utilization of terrestrial hydrological observations. In addition to the land surface models, LIS includes computational subsystems for DA, optimization and uncertainty estimation. LDT and LIS have been used to enable LDAS configurations over global (GLDAS; e.g., Rodell et al. 2004), North America (NLDAS; Mitchell et al., 2004; Xia et al., 2012) and regional (e.g., FEWS NET LDAS (FLDAS); McNally et al., 2017) domains. The development of LDT provides a formal environment to support the data synthesis requirements of the LIS-enabled LDAS instances. Specifically, LDT supports the processing of the model parameters, forcing data and initial conditions in a consistent manner, meets the DA related data preprocessing requirements,

climatological processing of data sets needed for model simulations, and the use of advanced data science techniques for data mining and fusion. The latest public release of LDT is version 7.2 and available at https://lis.gsfc.nasa.gov/releases.

The need for formal and efficient data fusion environments to augment modelling systems has been recognized in the "Model Data Fusion" (MDF; Raupach et al., 2005) paradigm, which describes the iterative nature of model development and the critical data dependencies and information transfer in the modelling process. The LIS framework has been designed to support this interplay between models and data through both internal and external components. The internal LIS subsystems for DA, optimization and uncertainty estimation allow the exploitation of the information from hydrological datasets for improving model structure, parameters and states. A post-processing environment known as the Land surface Verification Toolkit (LVT; Kumar et al., 2012) provides the capabilities for the verification, benchmarking, and evaluation of LIS and other independent model simulations and a wide range of observational datasets. Together with LIS and LVT, the development of LDT allows the capabilities for realizing the end-to-end MDF paradigm through formal environments that allow input data processing, mining and fusion, model characterization, formulation and validation.

This paper provides a detailed technical description of LDT, its capabilities and applications, highlighting its use as both stand-alone and within the overall LIS framework. Section 2 gives additional background and review of land model input processing software. Sections 3 and 4 describe LDT's overall design and variety of capabilities it currently supports. Several examples of some of the capabilities are provided in parts of Section 4. Finally, a summary and description of future work is contained in Section 5.

## 2. Background

There are a few instances of specialized data processing environments designed to support large modelling systems. One example includes the Community Land Model, versions 4 and higher (Oleson et al., 2010), which has data preprocessing scripts and online instructions provided to users to generate inputs for the model. The developers provide standardized global input files, but if the user wants to run for another resolution, regional subset or use different parameters (e.g., landcover map), the user must modify and run several different scripts to generate the necessary input files, which can take several steps. Other examples include the National Center for Atmospheric Research (NCAR) WRF Preprocessing System (WPS) and the pre-processor for the WRF Hydrological modelling extension (WRF-Hydro; Gochis et al., 2014; Sampson and Gochis, 2015). WPS offers a suite of specific datasets and primarily serves the preprocessing needs of the WRF community (Skamarock et al, 2008) and some in the Noah land surface model community (e.g., Chen et al., 2007). If the user wants to use WPS for Noah model parameter preprocessing, the user is either limited to what preprocessed parameters are available, or they have to generate those files to be in the specific WPS required format before using them. The WRF-Hydro preprocessor can utilize different hydrological-based topographical datasets, such as HydroSHEDS (Lehner et al., 2008), however the input elevation

maps to WRF-Hydro preprocessor are expected to be specifically in ArcGIS raster format, a proprietary format (ESRI, 2016), and may require more testing and effort when using open-source alternatives, like QGIS (http://qgis.osgeo.org).

Meteorological boundary conditions used in the numerical simulation of land surface states and fluxes are required, in many instances, to be downscaled and/or adjusted to the surface level as inputs to the land models. Some forcing data pre-processing efforts currently exist to downscale coarser scale datasets, e.g., climate model reanalyses in high varying terrain-based regions. Examples include the MERRA Spatial Downscaling for Hydrology tool (MSDH; Sen Gupta and Tarboton, 2016), which uses the R statistical software package (e.g., https://cran.r-project.org/); TopoSCALE, v.1.0 (Fiddes and Gruber, 2014); and the "eartH2Observe" data portal, which provides a suite of python scripts that downscale meteorological fields from the European Union's eartH2Observe dataset (https://github.com/earth2observe/downscaling-tools). However, these script-based or software toolkits typically only serve a select set of different meteorological forcing datasets.

LDT shares some commonality with these processing tools, but it is designed to be a more generic and comprehensive environment for supporting a wider range of data processing needs for the land and hydrological modelling communities. It provides the user many data processing options in how datasets get generated onto a common projection and grid, reducing inconsistencies and errors, especially when combining different parameter datasets. LDT uniquely supports the handling of a suite of land remote sensing measurements and preprocessing requirements for data assimilation environments. In addition to these key functionalities, LDT can generate certain model initial conditions (e.g., climatologically-averaged state fields) for deterministic and ensemble model runs, capability that is often needed in routine model simulations. Furthermore, the software is being enhanced with advanced techniques such as the development of data-driven models based on Machine Learning (ML) techniques and Bayesian merging for adaptive downscaling and bias-correction methods. LDT can handle input datasets in their "native" formats, performs consistency checks to ensure reasonable values (e.g., no missing values), and provides the outputs using the conventions and formats compliant with community data standards. Most processed outputs are written to a standardized, descriptive format known as the Network Common Data Format (NetCDF; Unidata, 2015).

## 3.  Software Design of the LDT Framework

As noted earlier, LDT is designed to encompass a broad set of functionalities that complement the modelling, data assimilation and evaluation environments of the LIS framework. Together, the LDT-LIS-LVT series conforms to the Model Data Fusion (MDF) concept (Raupach et al., 2005), where LDT supports the input data processing needs of the modelling system of LIS, and LVT provides the evaluation procedures to help with revising and improving any of the input and model formulations. Figure 1, modeled after the schematic outlined in Williams et al. (2009), highlights these end-to-end connections and capabilities in support of the MDF paradigm. LDT plays a central role in enabling this vision, by providing the data and

information processing capabilities, which LIS and LVT use to enable an iterative process of model formulation, state and parameter estimation and refinement, generalization, and model validation and benchmarking.

LDT shares a similar object-oriented framework design as LIS, with a number of points of flexibility known as "plugins". Specific implementations (such as soil parameter datasets, or a surface meteorological forcing) are added to the framework

through the plugin-interfaces. LDT uses the plugin-based architecture to support the processing of different types of observational data sets, ranging from in situ, satellite and remotely sensed and reanalysis products. The LDT software structure is organized into three layers: 1) the LDT core layer; 2) the "*Abstractions*" layer; and 3) the "*Use case*" layer represents the functional implementations of the Abstractions. Fig. 2 outlines this structure and what is defined further in each layer. The "*core*" top layer executes the generic functions of time management, defining the output fields, geospatial transforms, and top-

level handling of the different model parameters and meteorological dataset processing. The "*Abstractions*" layer enables "pluggable" interfaces with which to incorporate different features, run modes, model datasets, and other functionalities. Also, a key aspect of the *Abstractions* layer is the ability to reuse the plugins to support additional features and expand LDT's capabilities.

The LDT code is implemented in Fortran 90 and C programming languages. The C-language based virtual function table implementation is used to simulate polymorphic behavior for the extensible components in LDT. These function tables enable the "*Abstractions* layer" constructs. LDT is also supported by a variety of libraries, which handle not only the data format aspects (e.g., NetCDF I/O), but also the core routines as supplied by the Earth System Modelling Framework (ESMF; Hill et al., 2004) library. ESMF is a library framework to support the building and coupling of earth system model components.

ESMF provides several "off-the-shelf" infrastructure utilities such as clock/time manager and generic constructs for storing and exchanging data between various system components. LDT utilizes several ESMF features for passing information between the plugin components and the core routines.

A number of libraries to enable the support for common earth science data formats is also utilized in LDT. They include the

latest NetCDF, version 4 (NetCDF-4), Hierarchical Data Format (HDF5; The HDF Group, 2015), HDF-EOS (or HDF-4) and the GRidded Binary or General Regularly-distributed Information in Binary form (GRIB) data formats, versions 1 and 2. Currently, the GRIB data formats are supported using the European Centre for Medium-Range Weather Forecasts (ECMWF's)'s GRIB-API library (ECMWF, 20015) and will be replaced with the latest ECMWF's ecCodes. Finally, LDT handles other data format libraries, including Tagged Image File Format (TIFF) and Band Interleaved by Line (BIL) formats,

both used mostly with remotely sensed data and widely supported in GIS software environments and applications. TIFF formatted files are read in using the Geospatial Data Abstraction Library (GDAL; http://www.gdal.org/) translation library, which is linked and invoked via the FortranGIS project libraries (https://github.com/dcesari/fortrangis).

## 4. Capabilities and Features of LDT

LDT provides a range of features and capabilities that support the land surface and hydrological modelling communities. The current features and options are described further in detail below. Figure 3 provides an overview of the current LDT capabilities and components.

### 4.1 Model parameter processing support

For LSMs and hydrological models, the importance of providing representative or "realistic" physical parameters has been shown in several studies (e.g., Sun and Bosilovich, 1996; Duan et al., 2006; Bounoua et al., 2006; Nearing et al., 2016). The key parameter types required for LSMs include: (1) land cover/vegetation; (2) land/water mask; (3) soils; and (4) topography. Many land surface models contain tables of physical parameters that are indexed by spatial maps of parameter types (e.g., roughness length indexed by land cover type or saturated hydraulic conductivity indexed by soil texture class). Alternatively, physical parameters themselves may be specified on each model grid (e.g., snow free albedo; green vegetation fraction). Adjunct models to LSMs include streamflow routing models and lake models. These models may be included with or separate from the LSM. Depending on their dimensionality and complexity, streamflow routing models require information about flow directions, drainage areas, slopes, roughness, and lengths of river reaches. Similarly, lake models require information about lake area and depth(s).

The first major parameter type of any land-based model is the vegetation or land cover (or use) classification map. Not capturing the correct landcover at different scales can lead to errors or impacts on other modeled processes, e.g., coupled feedbacks (Bounoua et al., 2006). Another feature in some LSMs is the concept of representing subgrid heterogeneity, also referred to as subgrid "tiling". Instead of considering the dominant land characteristics only, the subgrid tiling approaches represent a gridcell as a mosaic of a number of homogeneous elements, determined from the distribution of land parameters within a grid cell (e.g., Avissar and Pielke, 1989; Koster and Suarez, 1992). Subgrid tiling is aimed at better representing land surface model effects and feedback to coupled atmospheric models (e.g., Giorgi and Avissar, 1997; Essery et al., 2003; de Vrese et al., 2016). In addition to vegetation-based tiling, the effects of soil moisture distribution (e.g., Entekhabi and Eagleson, 1989) and elevation-based subgrid variability (e.g., Leung and Ghan, 1995; Nijssen et al., 2001; Newman et al., 2014) on different water budget variables, like runoff, and atmospheric response have been investigated. LDT has been designed to support the representation of subgrid tiling for not only vegetation but also for multidimensional combinations of properties, including soil types and topographic derivatives (e.g., elevation, slope). Similar approaches have been developed for hydrological response units to capture subgrid heteorogeneity for land model processes (Chaney et al., 2016).

LDT uses the vegetation or land use map as a primary input parameter from which subgrid heterogeneity can be statistically represented and also a corresponding land-water mask can be derived. Figure 4 shows example vegetation tile frequency maps from four different vegetation classes (e.g., evergreen needle leaf, croplands) belonging to the Moderate Resolution Imaging Spectroradiometer (MODIS) International Geosphere-Biosphere Programme (IGBP) landcover classification map (Friedl et

al., 2002). LDT can read in a moderately high-resolution vegetation map (e.g., < 1 km per gridcell) and generate the tiled frequency maps, as highlighted in Figure 4. In addition to landcover, LDT also represents the subgrid scale distribution of soil types and topography datasets within a gridcell. The ability of LDT to represent the distribution of fine scale features of the underlying data for other land characteristics such as soils and topography allows a more flexible tiling representation, based on any of these features, or a combination of them. Landcover and land use map options in LDT include the U.S. Geological

Survey (USGS) 24-class landcover (USGS GLCC), the University of Maryland (UMD) Advanced Very-High Resolution Radiometer (AVHRR) landcover map (Hansen et al., 2000), and a few other dataset options, like Mosaic LSM vegetation types (Koster and Suarez, 1996) and JULES (Dunderale et al., 1999).

Closely related to the vegetation type and land use parameters described above is the "mask" field, which identifies valid grid

cells on which the model will run. Typically for a land surface or hydrological model, the mask discriminates between land and open water points, assigning an index value, like 1, to the valid land points. In LDT, such a mask can be derived from the land classification map or read in and imposed. If imposed, LDT ensures that the all processed parameters are geographically co-registered and consistent with the input mask. A variety of options exist in LDT to ensure consistency between the masks and model parameters. These options include allowing the user to select neighboring gridcells to "fill" in a parameter value

when the landmask indicates a valid land point, but the parameter has a missing value. If no valid neighboring values are available (e.g., in case for small islands), the user can then specify a universal value to fill the missing data. In addition, LDT offers other parameter processing features, such as upscaling (e.g., averaging) or downscaling techniques (e.g., bilinear interpolation), and different projections (e.g., equidistant geographic coordinate system, polar-stereographic).

Another key LSM parameter involves the representation of soil types. LDT offers a variety of data options, including soil texture and soil fraction-based maps (i.e., sand and clay fractions), depending on what the LSM needs. As mentioned above, sub-grid tiling can be generated and represented by LDT using both the soil texture and soil-fraction maps. Some of the current soil data options include the original Food and Agricultural Organization soil texture and fraction maps (Reynolds et al., 2000), the blended STATSGO (Miller and White, 1998), version 1, and the Food and Agriculture Organization (FAO) global soil

texture map (Reynolds et al., 2000), and the International Soil Reference and Information Centre (ISRIC) texture, fractional and other soil property-based dataset (Hengl et al., 2014). Figure 5 shows an example comparison of soil texture classes over the U.S. from the STATSGO-FAO and ISRIC soil texture maps, as processed through LDT.

Currently, processing of the parameters for several land surface and hydrologic models are supported by LDT (and LIS), as summarized in Table 1. These models are the state-of-the art in representing the key processes of the terrestrial energy, water and carbon cycles as well as specialized process representations of specific features of the land surface (e.g., lakes, urban). Some of the LSMs include Noah versions 2.7.1 and later (Chen et al, 1996), Catchment LSM (Koster et al., 2000), JULES (Best et al., 2011), and several others. LDT also processes final inputs for the Hydrological Modelling and Analysis Platform (HyMAP) (Getirana et al., 2012; Getirana et al., 2017), which is a hydrological routing scheme in LIS and collects and routes LSM-based total runoff through a network of catchments and tributaries to major river stems. Finally, LDT supports the processing of lake model parameters, e.g., water depth for freshwater lake models such as FLake (Mironov et al., 2006).

The complexity of these model formulations continues to increase with the addition of new components (e.g., crop, groundwater models), fine scale modelling needs (e.g., topographical downscaling) and efforts to include impacts of human management (e.g., irrigation). LDT provides a number of schemes and datasets to address the data processing requirements of these additional components. For example, the processing of irrigation intensity information from the MODIS-based irrigation map developed by Ozdogan and Gutman (2008) and the Global Rain-Fed, Irrigated and Paddy Croplands (GRIPC; Salmon et al., 2015) are supported within LDT. Also, crop information is available, which includes the "CROPMAP" scheme in Ozdogan et al. (2010), and the updated, high-variety crop map of Monfreda et al. (2008).

To enable the topographical downscaling of meteorological fields for fine scale modelling, LDT processes elevation, slope and aspect datasets. High resolution precipitation climatology maps from the Parameter-elevation Relationships on Independent Slopes Model (PRISM; Daly et al., 1997) or from WorldClim (Fick and Hijmans, 2017) can be ingested within LDT for downscaling and bias-correcting precipitation fields. LDT also supports different topographic map options (e.g., elevation, slope), which include the GTOPO30 (Gesch et al., 1999) and the Shuttle Radar Topography Mission, 30 arc second (SRTM30; Jarvis et al., 2008) digital elevation model (DEM) datasets (globally, 30 arc second resolution versions).

## 4.2 Generation of model initial conditions

Similar to model parameters, model initial conditions (ICs) are required by all LSMs to simulate land surface model states and fluxes (e.g., Cosgrove et al., 2004; Rodell et al., 2005). Climatologically averaged, state-based initial conditions have been shown to provide more optimal initial conditions for LSM and hydrological model simulations than other methods (Rodell et al., 2005). One example of improving the model initial conditions was shown in Xia et al. (2012) where going from a 1-year spin-up period, originally used in the North American LDAS, phase 1 (NLDAS-1), to two-stages of running several years and averaging selected dates (e.g., Jan. 1) for NLDAS, phase 2 (NLDAS-2). Running for several years improved the initial conditions for the NLDAS-2 model simulations, whereas the 1-year NLDAS-1 spin-up produced "lingering effects" on the soil moisture fields. LDT offers a feature to generate such climatological initial conditions. The climatological initial

conditions are generated by taking an average of the same date and time (e.g., June 1, at 00Z) over multiple years (e.g., 1982-2010).

LDT also provides the capability to produce an IC-based file to initialize an ensemble simulation, e.g., for a seasonal forecast
ensemble, going from a single member model "restart" file to a multi-member file, which we refer to as ensemble "disaggregation". In addition, an option exists to calculate the ensemble average from a multi-member IC (or restart) file to form a single-member IC file, which we refer to as ensemble "aggregation". These options can support initializing data assimilation and forecast ensemble model simulations.

**4.3 Data processing support for land data assimilation**

The use of observational data from satellites and other remote sensing platforms is a growing area of research in the land/hydrological modelling community. The information from these observational data sources are often used to improve the characterization of models states through data assimilation (DA; e.g., Reichle et al., 2002; Kumar et al., 2008c) and model parameters through inverse modelling techniques (e.g., Harrison et al., 2012). The computational systems of DA and inverse
modelling, built around the physical models, have their own data and processing requirements. Most DA systems are designed to address and improve the random errors in models and expect the input datasets to be generally unbiased relative to model estimates. A common approach in the land DA community to enable these "bias-blind" (Dee, 2005) systems is to rescale the observational data to be consistent with the model climatology, which is simply a multi-year average of model states. The development of model and observational data climatologies to enable such reprocessing is supported within LDT.

For soil moisture data assimilation, a commonly used rescaling approach is called "CDF-matching" where cumulative distribution functions (CDFs) are used to bias-correct and reduce differences in observation and model states (Reichle and Koster, 2004). This scaling approach matches the CDF of the observation to that of the model and corrects all moments (e.g., first and second) of the observation distribution, regardless of its shape. To generate CDFs with LDT, the user must supply
multiple years of model output and observational data for the a given variable. LDT then produces model- and observation-based CDF data, separately, at each model grid point, which the DA system can use to perform the rescaling. The user can select the granularity, temporal averaging period and data count threshold to generate the CDF files. The CDFs can also be generated either based on lumped annual-based values ("lumped") or seasonally stratified CDF values (i.e., "monthly"). Kumar et al. (2015) demonstrated that the use of seasonal CDFs reduces the statistical errors from CDF-matching in soil
moisture DA, compared to the use of lumped CDFs. Finally, LDT can account for spatial sampling by using neighboring pixels to increase the sampling density in the CDF calculations (e.g., when a data record period is short; based on Reichle and Koster, 2004), or by grouping CDFs by landcover or soil texture type.

LDT supports several different satellite-based observational data types that can be used for data assimilation in LIS. These satellite-based observations include a variety of soil moisture (SM) retrievals, terrestrial water storage (TWS), and snow depth (SNWD). Table 2 summarizes the various products available in LDT, which encapsulate most modern land remote sensing measurements.

The NASA's Gravity Recovery and Climate Experiment (GRACE) TWS anomaly dataset is part of the suite of satellite products that can be processed by LDT and assimilated into LIS. Currently, LDT supports monthly gridded GRACE mass products either in 0.5° or 1.0° resolution, regardless of their processing methods (i.e., spherical harmonic or mass concentration). Release-05, or "RL05" products have been provided by University of Texas Center for Space Research (CSR),

NASA's Jet Propulsion Laboratory (JPL), and the German Research Centre for Geosciences (GFZ). Along with the GRACE anomaly product, LDT can incorporate GRACE scaling information and leakage errors that are provided with the spherical harmonic products (Kumar et al., 2016). In addition, CSR provides a higher resolution RL05 version at 0.5 degree but using the Mascon solution (Save et al., 2016). LDT reads in the raw GRACE anomaly data and incorporates that information with model-based TWS information (units of mm), for example, as in Kumar et al. (2016). The final data produced are referred to

as "total TWS". Figure 6 shows an example of the LDT-produced total TWS, after incorporating the GRACE TWS anomaly information.

LDT also allows the definition of an "exchange grid" for DA, a domain that is used for the calculation of the observation minus the model forecast estimates (called 'innovations'). The use of the exchange grid allows improved consistency between

observations and the simulated model forecasts. The exchange grid information generated by LDT is then employed by the DA system in the calculation of data assimilation updates.

## 4.4 Processing support for meteorological forcing datasets

LSMs driven with higher spatial resolution and observational data have been shown to have improved land states and fluxes

over coarser and model-only generated meteorological inputs (e.g., Masson et al., 2003; Reichle et al., 2011). Higher resolution forcings can improve land model ICs, for example, in coupled atmospheric simulations (e.g., Kumar et al, 2008a; Case et al., 2008). LIS and LDT support a large variety of meteorological reanalysis, observational forcings and seasonal climate forecast datasets. LDT supports a large suite of meteorological forcing data, and it can be used as a stand-alone tool to downscale spatially and temporally, merge and quality control these different forcing datasets. The final meteorological fields are written

to a single file in NetCDF-4 format.

At this time, LDT supports two basic ways of processing meteorological datasets. First, LDT can be used to spatially and temporally interpolate to downscale and merge (or "overlay") different meteorological forcing datasets using the "Metforcing

processing" run mode option. A second option exists where the user can generate climatological forcing datasets to capture diurnal and seasonal cycles of longer term forcing data records. This second feature works with a variety of meteorological datasets, including overlaying multiple datasets, to generate a more comprehensive climatological forcing (available down to an hourly climatology). This climatology option can be used for different applications, including generating forcing used in forcing ensembles and climatology forecast capabilities.

## 4.5 Spatial and temporal forcing downscaling and disaggregation options

Generating higher resolution meteorological inputs can be very important to driving the LSM or hydrological models, especially over mountainous regions, to better capture fine-scale features, such as variations in temperature or incoming solar radiation on snowpack dynamics (e.g., Rasmussen et al., 2011). Several studies have further downscaled reanalysis and forecast datasets, which include seasonal climate and climate change (e.g., Maraun et al., 2010), showing improved meteorological and hydrological representation at those scales. Also, temporal disaggregation of coarser timescale forcing data (e.g., daily) has been shown to improve hydrological representation versus simply applying a uniform rate (e.g., over the day; Ryo et al., 2014). Such methods are also applied in GLDAS (Rodell et al., 2004) and NLDAS (Cosgrove et al., 2004) forcing downscaling approaches.

LDT offers some options for either spatially or temporally disaggregating forcing datasets. For temporal disaggregation, forcing datasets that are at coarser timesteps, e.g., daily or greater, can be interpolated to a finer timestep (e.g., 3-hourly). For example, daily observed precipitation fields can be disaggregated using precipitation fields at finer timesteps, e.g., hourly fields from the Modern-Era Retrospective analysis for Research and Applications, version 2 (MERRA-2), by applying weights from the MERRA-2 precipitation to create sub-daily precipitation from the daily product. This approach is based on Cosgrove et al. (2003), and it is preferred for LSMs over other methods, e.g. simply distributing a daily precipitation product at the same rate (uniform) over each subdaily (e.g., 3-hourly) timestep (e.g., Sen Gupta and Tarboton, 2016).

Current spatial downscaling techniques available from LDT, in conjunction with LIS, include using higher resolution (e.g., 1 km), monthly precipitation climatology datasets, such as from the PRISM (Daly et al., 1997) or WorldClim (Fick and Hijmans, 2017) to spatially downscale coarser-scale precipitation data. Specifically, LDT calculates and stores the ratio of high-resolution precipitation climatology versus the same climatology aggregated at the coarser-scale resolution. These ratios reflect how spatial patterns of monthly precipitation change with respect to spatial resolutions and therefore provide a basis for spatially downscaling precipitation data when read into LIS. If the climatology of the precipitation data used to run LIS is also available, spatial downscaling can be performed in conjunction with bias-correction. In this case, for example, LDT calculates the ratio of 1 km PRISM climatology to that of the coarser-scale precipitation used by LIS and stores the ratio (at the simulation resolution) in the LIS parameter file. LIS in turn reads the ratio and applies it to precipitation data each time

when new forcing data are read.  By definition, the output precipitation field from LIS will have the same climatology as PRISM in each calendar month, hence removing the bias of the coarser-scale precipitation climatology relative to that of the finer-scale precipitation climatology.

Other spatial disaggregation techniques available in LDT include the ability to process topographic maps (e.g., elevation, slope) and forcing-based lowest layer terrain heights, which can be used in LIS to further downscale the forcing fields in two different ways.  The first approach follows that used in NLDAS-1 and 2 (e.g., Cosgrove et al., 2003), where a static environmental lapse rate (of 6.5 K / km) is used to apply an elevation adjustment to the spatially coarser meteorological fields (e.g., air temperature, specific humidity) to finer scales (e.g., 1 km) to capture greater terrain spatial variability.  This lapse-

rate correction can improve, for instance, air temperature representation in mountainous regions.  Figure 7 highlights the comparison of NLDAS-2 forcing dataset at its native 12.5 km (or 0.125 degree) resolution and then downscaled using the lapse-rate adjustment method with SRTM elevation to 1 km resolution using the SRTM 1 km elevation parameter file.  The ability to generate spatially-varying atmospheric lapse-rates based on atmospheric pressure levels and temperatures (e.g., Sen Gupta and Tarboton, 2016), is not available within LDT, but it could be expanded to include this approach.  Finally, LDT

processes and provides high resolution slope and aspect fields, which get applied in LIS to adjust downward solar radiation fields.  Accounting for slope and aspect has been shown to improve radiation budgets and snow simulations in mountainous regions (e.g., Kumar et al., 2013).

### 4.6  The Machine Learning Layer

Despite the huge advancements in modelling made possible by "physical" models, they have fundamental limitations in their ability to accurately portray the complex processes of the Earth system. For example, the significant human footprint on the hydrological cycle has essentially led to a "replumbing" of the global hydrological cycle through activities such as agriculture and infrastructure development, leading to the recognition of a new geological epoch called the Anthropocene (Zalasiewicz et al. 2011). The accurate representation of the replumbing is critical for understanding the consequences of human activity on

water resources and its contribution to hydrological extremes. Due to the often subjective nature of the human-engineered processes, the conceptual physical models are limited in their ability to represent Anthropocene processes. On the other hand, large scale observations from satellites and remote sensing platforms provide a huge opportunity to represent them, which is only possible if sophisticated data processing and data driven models are available to fully exploit the information content of such measurements.

The availability of increased amounts of earth science data and the power of modern computers presents an ideal scenario for employing machine learning (ML) techniques for data driven modelling and predictive analytics. ML-methods essentially develop non-linear feature transformations learned from mapping a set of inputs to a set of outputs. More recent advancements

in ML such as deep learning (Bengio, 2009), modeled after the human cognitive process, allow the modelling of more complex relationships among the data and incremental learning. Generally, the data driven ML-models are a good alternative to the physical models when it is difficult to build knowledge-driven simulation models in cases where the understanding of the underlying processes is lacking. With this recognition, LDT includes a ML layer designed to support a variety of ML algorithms and training models. The ML models developed from LDT is expected to augment the physical models and data assimilation environments.

Currently the ML-layer in LDT includes shallow learning algorithms such as Artificial Neural Network (ANN), which consists of an input layer, output layer and a set of hidden layers. The user specifies the input and output layers, whereas the topology of the hidden layer is constructed within LDT. During the training phase, LDT is presented with a set of inputs and the corresponding outputs, which are used to develop a set of weights and interconnections within the ANN. The trained network can then be used for generating predictions with a new set of inputs.

The ML-based trained network models can be a useful operator within DA environments. Most satellite instruments detect radiances (electromagnetic energy over specific wavelengths) and conversion of these raw measurements to geophysical variable is not always trivial. The ML techniques can be used to develop models that translate between radiance measurements and related geophysical quantities. Such models can then be used in DA configurations, essentially allowing the direct use of raw satellite measurements in modelling.

An example of such a scenario is presented below in Figure 8. The input ML layer in LDT is used to ingest radiance measurements for the 18 and 36 GHz channels (both horizontal and vertical polarizations) from the AMSR2 instrument on the Global Change Observation Mission-Water (GCOM-W) satellite (Wentz et al. (2014). In addition, the input layer is presented with the fractional snow cover data from MODIS Terra instrument (MOD10A1) and the outputs from a LIS model simulation (variables including precipitation, green vegetation fraction, soil moisture and soil temperature). The ANN within LDT is then trained against the daily snow depth measurements from the Global Historical Climate Network (GHCN) for a period of approximately year (1 Aug, 2012 to 31 July, 2013). The training is conducted at a point location (Tierra Amarilla in New Mexico, 36.7°E, 106.6°W), where the snow evolution is often ephemeral, making the accurate prediction difficult. The bottom panel of Figure 8 shows the performance of the trained network, when used for prediction in the following year (1 Aug 2013 to 31 July 2014). The snow evolution is captured well by the ANN-based predictions. The specification of the input and output layers is user-defined and customizable. The ML layer in LDT can also be used for developing data-driven models both in a spatially distributed manner (where the training is done on a gridcell by gricell basis) and on an aggregate basis (where a single trained model is developed using available inputs for all gridcells).

## 5. Summary and future capabilities

Land Data Assimilation Systems (LDASs) require the integration of high quality observations with state-of-the-art land surface and hydrological models to acquire robust estimates of land surface conditions to meet the needs of applications involving weather and climate modelling, water resources management, modelling of hydrological extremes, among others. The synthesis of several types of model and observation data across various spatial and temporal resolutions and extents is needed to support the development of flexible LDAS configurations for conducting both research and application-oriented studies. To offer such a data fusion software framework, the Land surface Data Toolkit (LDT) has been developed with a large suite of capabilities including: (1) parameter processing for a wide variety of models including land surface, hydrological, lake and streamflow models; (2) to create initial conditions (e.g., climatological restarts) from model runs; (3) data assimilation preprocessing support; (4) meteorological forcing data processing for inputs to the models; and (5) data driven models based on machine learning to assist the physical modelling and DA environments. LDT provides a formal environment to handle the data related needs within the model-data-fusion concept, which is recognized to be essential for the systematic development and improvement of Earth system models.

LDT serves as the main preprocessor to the NASA LIS, which is an integrated framework designed for multi-model LSM and DA integrations. LDT can also be used independently as an observational and model input processor for other land surface modelling systems. In addition, LDT offers a variety of user options to process model inputs, supports a variety of software libraries, the ability to read in "native" (or original) dataset formats, and uses common data formats, like NetCDF-4.

LDT is an evolving framework and will continue to be developed with the addition of support for new datasets and data processing algorithms. Over the past several decades, the complexity of land surface models has gradually increased, as they have evolved from the first-generation simple bucket schemes (Manabe, 1969) to models that represent the complex interactions of the terrestrial water, energy and biogeochemical cycles. In addition, more detailed and fine-scale representations of the land surface (surface and sub-surface) also continue to grow, imposing an increased set of data requirements for their effective applications at the scales of interest. A formal, extensive and adaptive environment such as LDT is necessary to support these requirements. Similarly, the land DA applications and their complexity continue to grow with the increasing availability of remote sensing observations. Sophisticated data fusion models and processing algorithms are required to support the utilization of raw satellite measurements. With the increase in computing power and data science advancements, the machine learning and predictive analytics have become more commonplace in areas involving e-commerce, social media and healthcare. The data-rich Earth science arena is an ideal environment for deploying such data science enhancements and the ML layers in LDT will be continually updated to exploit such capabilities.

Future LDT capabilities will continue to include new parameter datasets (e.g., landcover, soil types), remotely sensed and in situ observations for DA preprocessing needs, projection grid types (e.g., Mercator projection), additional meteorological forcing datasets and downscaling techniques, and additional machine-learning methods. Parallel decomposition ability is also being developed and supported in LDT. Currently parallel capability is being tested with the meteorological forcing processing

and downscaling and with some of the parameter processing features. New LSMs are currently being implemented, including the Community Land Model (CLM; Oleson et al., 2010), and the latest versions of Noah and JULES LSMs. In addition, "native" parameter processing support is being considered for the Catchment and VIC LSMs, HyMAP and for other groundwater-based parameters. Finally, end-to-end data input processing for optimization and parameter estimation along with uncertainty estimation techniques have been considered in future LDT versions, another component fulfilling the MDF

paradigm with LIS and LVT.

## 6.   Code availability

The current version of LDT is Release 7.2r version (6 May, 2017 release), which is open source and publicly available from the main LIS website at https://lis.gsfc.nasa.gov/releases. The persistent identifier for this version is http://doi.org/10.5281/zenodo.1322613. The main LDT features described in this paper can be found with this release. Also,

end-use test case examples are provided (https://lis.gsfc.nasa.gov/tests/ldt) and additional documentation, including the full users' guide and tutorial type presentations, is located here (https://lis.gsfc.nasa.gov/documentation/ldt). Future versions of the code will be made also available on GitHub (https://github.com/).

## Competing Interests Statement

The authors declare that they have no conflict of interest.

## Acknowledgements

We gratefully acknowledge the financial support from NASA Earth Science Technology Office (ESTO), the Air Force Life Cycle Management Center, NASA Applied Sciences Water Resources Program, NOAA's Climate Program Office (MAPP program), NASA's High Mountain Asia, SERVIR Applied Sciences Team and Terrestrial Hydrology, NASA's Internal Research And Development (IRAD) program and NASA's National Climate Assessment (NCA) program, which all

contributed to the development of LDT. Computing was supported by the resources at the NASA Center for Climate Simulation (NCCS). We also want to thank our user community for their invaluable feedback in supporting LDT's development and feature implementation.

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

30  **Table 1. LSMs and some of their parameters supported in LDT. Note that several LSMs use the land cover type and/or the soil texture type for each tile within LIS in combination with a look-up table to generate vegetation or soil parameters for that tile.**

| Model | Type | Parameters | References |
|---|---|---|---|
| Noah, versions 2.7.1 and greater | LSM | MODIS-IGBP landcover, soil texture, monthly climatological greenness fraction and albedo, max. snow albedo, bottom soil profile temperature, slope type. | Chen et al. (1996) |
| Noah-MP, version 3.6.1 | LSM | Same as used in Noah LSM versions | Niu et al. (2011) |
| Catchment | LSM | Mosaic-based land cover classes, soil parameters (porosity, saturated hydraulic conductivity, Clapp-Hornberger PSI and B parameters), bedrock depth, wetness/shape/baseflow/water transfer/minimum theta/topographic tau parameters, diffuse and direct NIR/VIS albedo scale factors, monthly climatological greenness fraction and LAI | Koster et al. (2000a, b) |
| Mosaic | LSM | Soil sand/silt/clay fractions, soil porosity and color, monthly SAI and LAI maps. | Koster and Saurez (1996) |
| Simple Biosphere, version 2 (SiB-2) | LSM | UMD-based landcover, vegetation canopy parameters, rooting depth, leaf characteristics (e.g., photosynthesis, stomatal conductance), soil respiration, etc. | Sellers et al. (1996) |
| SAC-HTET, Snow-17 | LSM | SAC: Soil parameters (e.g., max. water storage, free water depletion rate), PET monthly maps, greenness vegetation fraction, snow albedo. | Koren et al. (2010) |
| Rapid Update Cycle (RUC) LSM v3.7 | LSM | Same parameters as Noah LSM, but also LAI monthly climatology. | Smirnova et al. (2016) |

| | | | |
|---|---|---|---|
| Variable Infiltration Capacity (VIC) v4.x | LSM | UMD landcover and landmask | Liang et al. (1994) |
| GeoWRSI, v2 | LSM | Start-of-season climatology, end-of-season climatology, length of growing period, and soil water content. | Verdin and Klaver (2002) |
| Community Atmosphere Biosphere Land Exchange (CABLE) model | LSM | Soil fractions and texture, porosity, landcover classification map | Kowalczyk et al. (2013) |
| Joint UK Land Environment Simulator (JULES), v4.3 | LSM | UM/JULES 10-km plant functional type (PFT) map, soil hydrology parameters (e.g., porosity, wilting point, saturated water conductivity, thermal capacity, thermal conductivity, and ground albedo). | Best et al. (2011) |
| Hydrological Modelling and Analysis Platform (HyMAP) v1, v2 | Routing | X,Y flow direction components, flood height, baseflow, basin domains and mask, runoff delay terms. grid elevation, river dimensions, e.g., height, length, etc. | Getirana et al. (2012, 2017) |
| Freshwater Lake (FLake) model | Lake | Interior water and lake depth, water-body quality-control information, lake wind fetch, lake sediment inputs | Mironov et al. (2006) |

**Table 2.  Different DA remotely-sensed observational or land-surface model data types supported in LDT and LIS.**

| Dataset type | Description | Reference |
|---|---|---|
| LPRM AMSR-E SM | The Land Parameter Retrieval Model (LPRM)'s Advanced Microwave Scanning Radiometer-Earth Observing System (AMSR-E) soil moisture retrievals. | Owe et al. (2008) |
| WindSat SM | WindSat passive microwave soil moisture | Li et al. (2010) |
| TUW ASCAT SM | ESA's Advanced Scatterometer (ASCAT) soil moisture, processed at Technische Universitat Wien, Netherlands. | Bartalis et al. (2008) |
| SMOS SM | ESA's Soil Moisture Ocean Salinity (SMOS) soil moisture dataset | Kerr et al. (2001) |
| GCOM-W AMSR2 SM | Global Change Observation Mission (GCOM) AMSR version 2 soil moisture | Wentz et al. (2014) |
| SMAP SM | NASA's Soil Moisture Active-Passive (SMAP) level 3 soil moisture products. | Entekhabi et al. (2014) |
| SMOPS SM | NOAA's Soil Moisture Operational Product Systems (SMOPS), includes several soil moisture datasets: AMSR2, SMOS, and ASCAT | Liu et al. (2016) |
| ESA's CCI ECV active+passive SM | ESA's Climate Change Initiative (CCI) Essential climate variable (ECV) blended active / passive microwave SM | Liu et al. (2011) |
| GRACE TWS | NASA's Gravity Recovery and Climate Experiment (GRACE) TWS anomaly dataset | Tapley et al. (2004) |
| GCOMW AMSR2 SNWD | AMSR2 passive microwave snow depth retrievals | Wentz et al. (2014) |
| LIS LSM model output | LIS land surface model output fields (e.g., soil | Kumar et al. (2008a, 2008b) |

| | moisture) | |
|---|---|---|

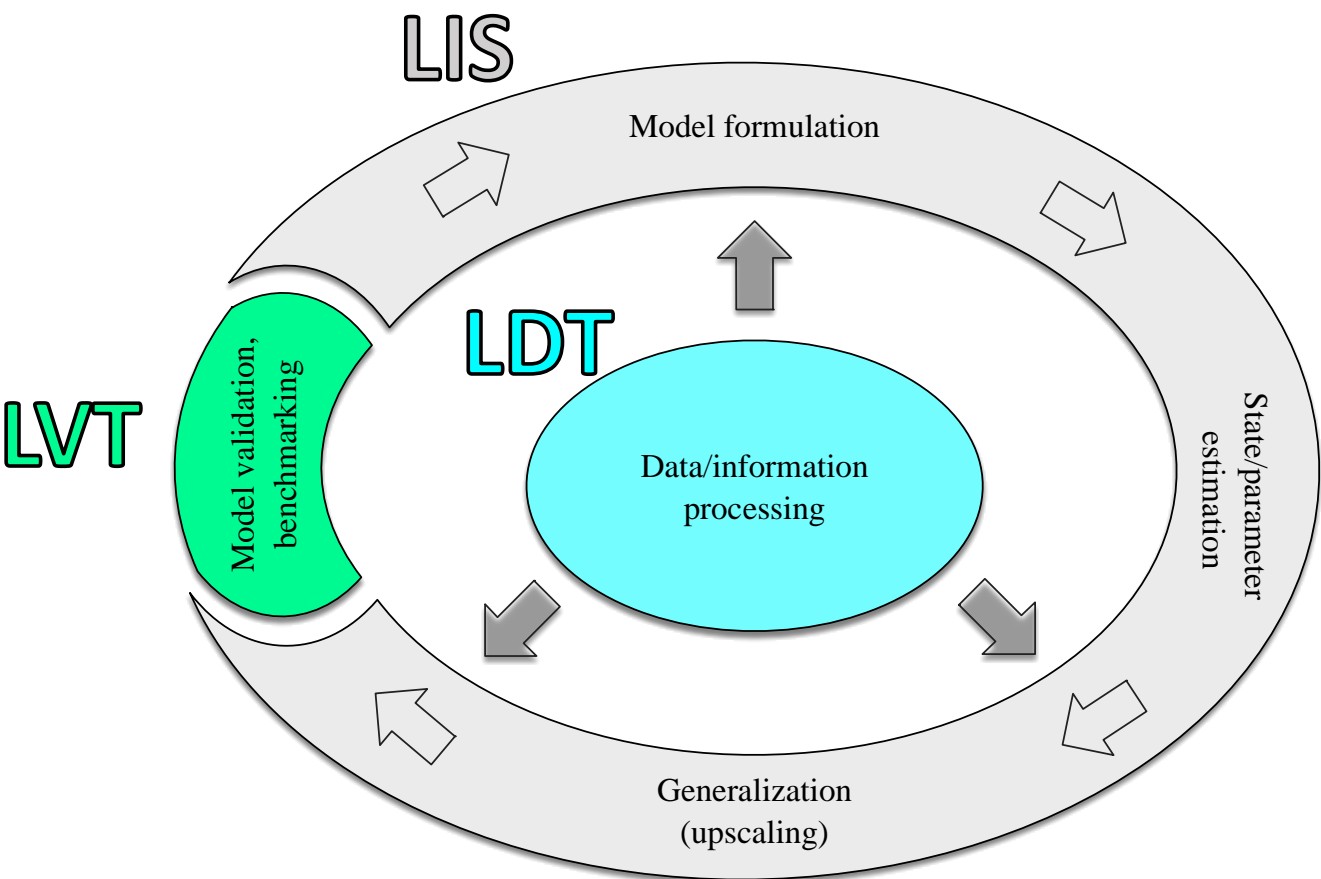

**Figure 1: Schematic of the complete model data fusion (MDF) paradigm enabled by LDT, LIS and LVT (modeled after the Figure 1 in Williams et al., 2009). LDT is the data preprocessing environment that feeds into the modelling and data assimilation environment of LIS, and also LVT, the model evaluation and benchmarking system.**

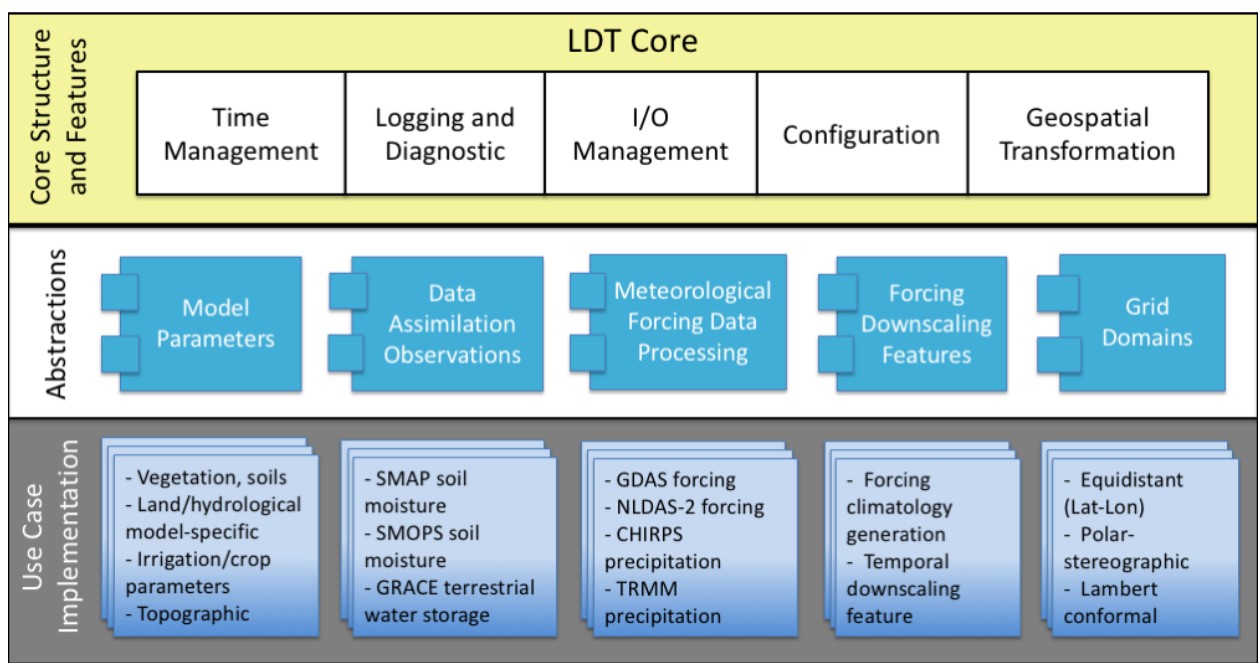

**Figure 2.** Schematic of LDT's main software architecture, showing the various core structures, abstraction layer, and use case implementations.

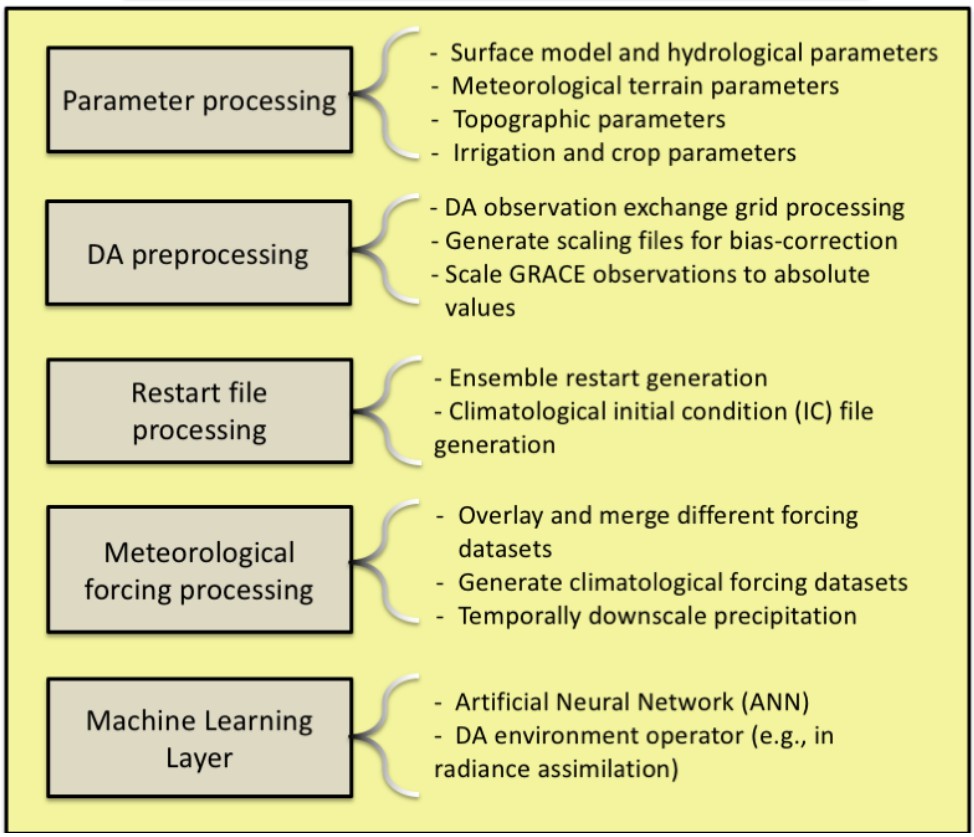

**Figure 3. Schematic depicting the current and different components in LDT.**

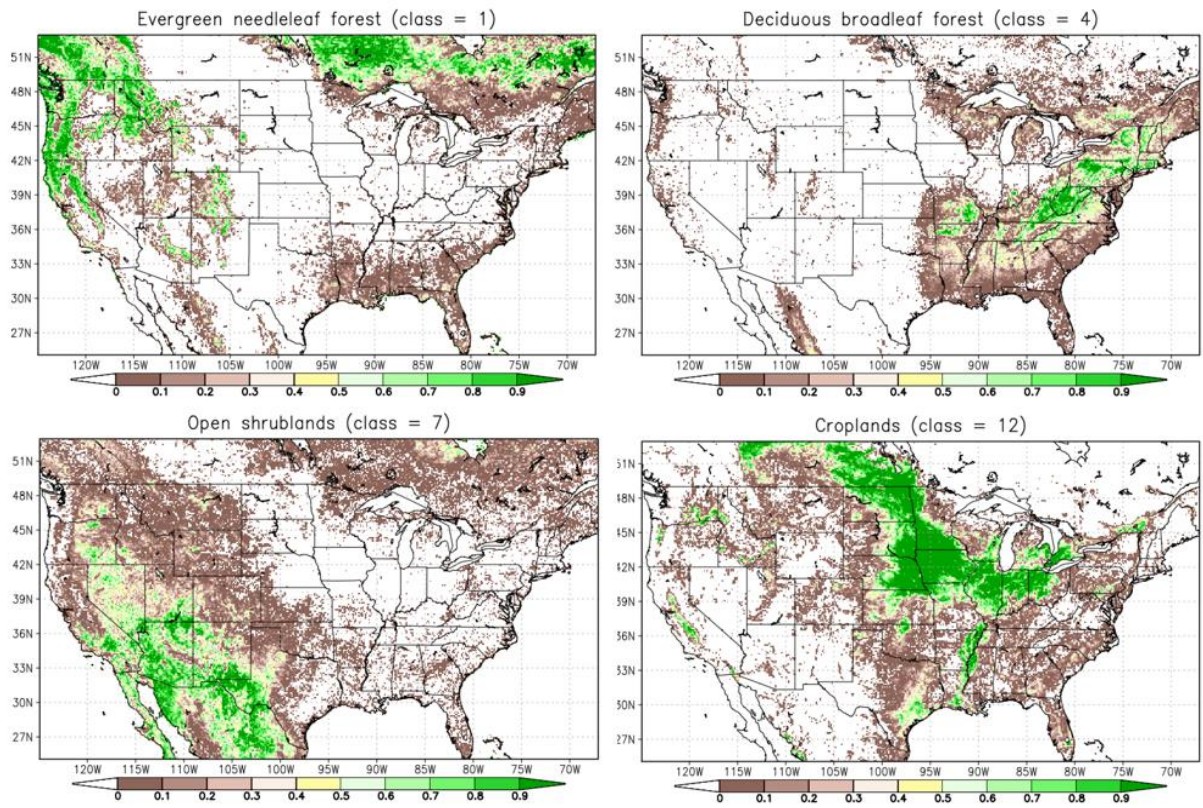

**Figure 4.** Vegetation distribution fraction of four different MODIS IGBP land cover classes (as produced by LDT): evergreen needleleaf (upper-left), deciduous broadleaf forest (upper-right), open shrublands (lower-left), and general cropland (lower-right). Values greater than 0.9 indicate where more than 90% of given gridcell (0.125° gridcell resolution, in this example) is dominated by that vegetation type.

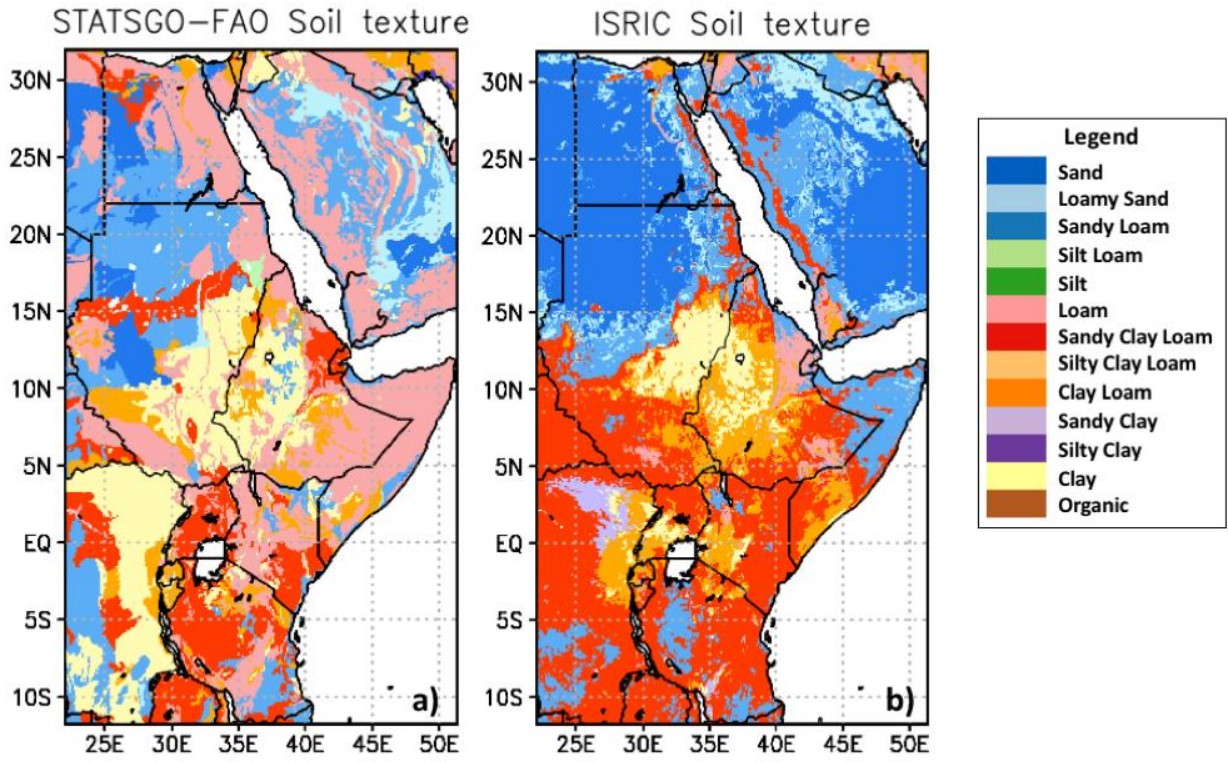

**Figure 5.** Comparison of the a) STATSGO-FAO soil texture class map (originally at 1 km resolution) versus the b) ISRIC soil texture map (originally at 250 m resolution). Dominant texture classes are shown here at 10 km spatial resolution.

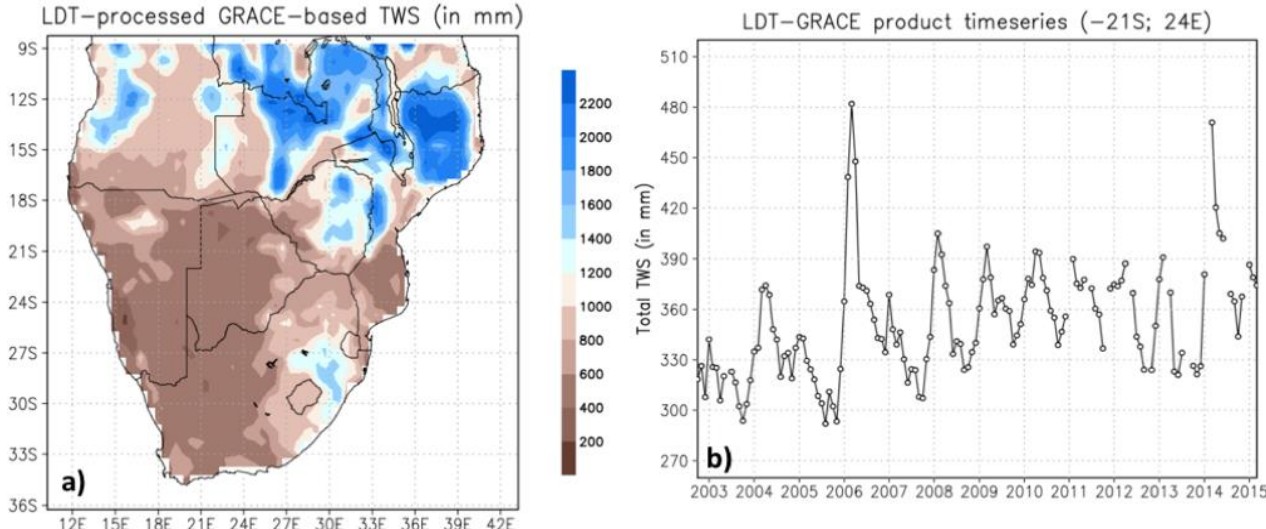

**Figure 6. Examples of LDT-processed GRACE-based total TWS (in mm). a) Plot of the LDT-processed TWS data for the Southern Africa region for February, 2011, and b) a time-series plot of the TWS data for years 2003-2015 and latitude of -21°S and longitude of 24°E.**

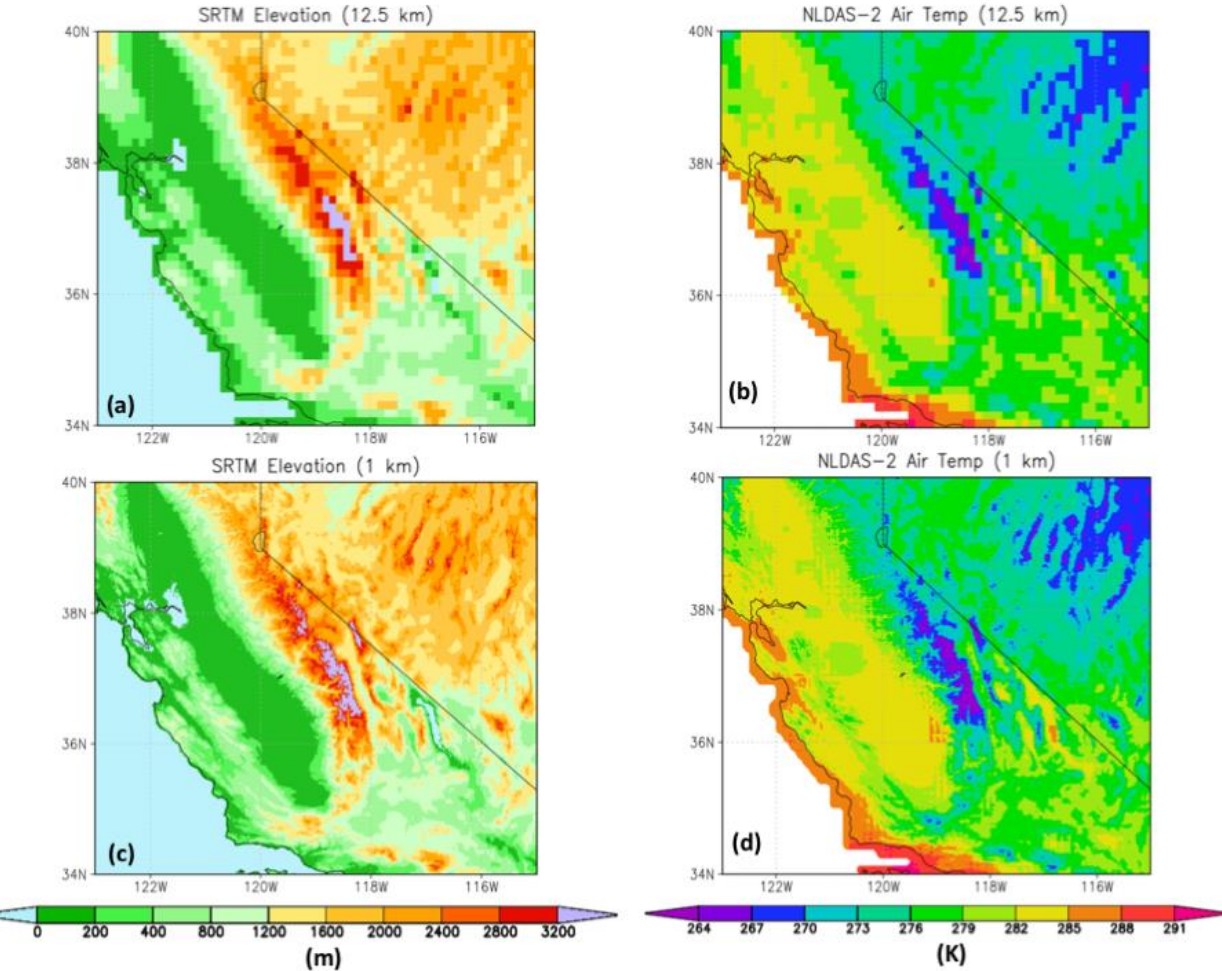

**Figure 7.   Examples of LDT-processed SRTM elevation parameter (in meters) at both a) 12.5 km and c) 1 km resolutions.  b) NLDAS-2 air temperature forcing field at its "native" 12.5 km resolution on April 1, 2005 (18Z).  d) The 1 km resolution SRTM elevation field was then used to "topographically downscale" the NLDAS-2 air temperature (in unit of K) using the lapse-rate correction approach for the finer detailed 1 km air temperature field, as shown in plot d).**

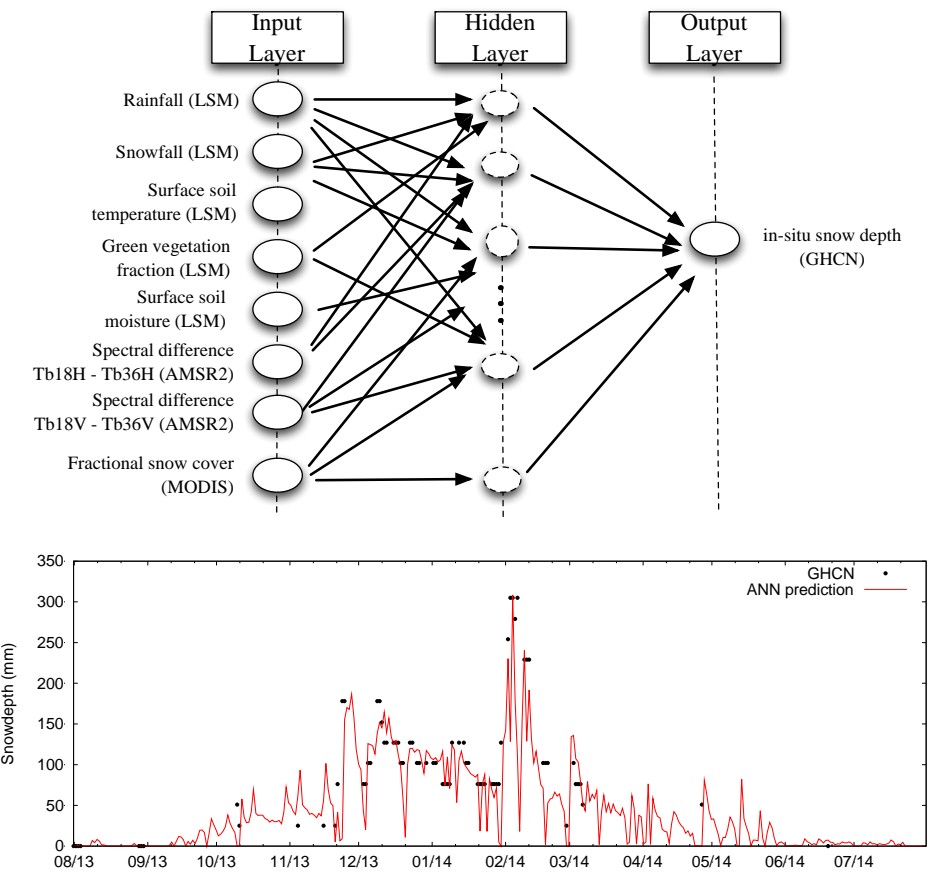

**Figure 8. Example of the ML layer utilization in LDT. The top panel shows the schematic of the ANN which ingests a suite of LSM-based and remote sensing-based inputs for developing predictions of snow depth. The bottom panel shows the performance of the trained network against in situ observations from the GHCN network.**