# Peer review of "The Land surface Data Toolkit (LDT v7.2) – a data fusion environment for land data assimilation systems"

_Geoscientific Model Development, 2018_

## Short Comment (SC1) · 4 May 2018

following the GMD guide line on code availability in https://www.geoscientific-model-development.net/about/manuscript_types.html the authors are encouraged to provide a DOI to the model release, for instance by via zenodo.org.

Lutz Gross GMD Executive Editor

---

## Referee Comment (RC1) · Anonymous Referee #1 · 23 May 2018

The authors have described a specialized land model processing tool, the NASA Land Information System Data Toolkit (LDT). This tool has been developed to support applications of the Land Information System, so it is more usefully viewed as a modeling system specific pre-processor than as a software tool for general use by the hydrological community. Nevertheless, LDT should be of interest to journal readers both because LIS and its supported Land Surface Models are widely used by the community and because the classes of processing capabilities included in LDT are of general interest for many advanced hydrological modeling systems.

The manuscript appears to provide an accurate description of LDT, and the level of

detail is adequate for the reader to understand the motivation and capabilities of the tool. I am happy to recommend the manuscript for final publication without further modification.

---

## Author Comment (AC1) · 16 Jun 2018

Thank you for this suggestion. We can have a DOI in the final paper version.

---

## Author Comment (AC2) · 16 Jun 2018

Thank you for your thorough review of our manuscript and for the positive feedback. We appreciate your comments.

We just want to quickly note that we do develop LDT not only to support the Land Information Systems (LIS) software but also the broader land modeling and hydrology community.

---

## Referee Comment (RC2) · Anonymous Referee #2 · 29 Jun 2018

This paper presents an overview of the land surface data toolkit v7.2. The main purpose of the paper is to explain the functionality of the toolkit and present the justification for various components with reference to the literature. Such tools are essential to the community as a means of facilitating model development and implementation, especially given the ever-increasing availability of observational data and computing resources.

The paper is not a technical description of the toolkit as such, more a very detailed description of the components. I was a little sceptical about the value of the paper for this reason, however after reading the content I believe it to be a worthy contribution

to the literature and reference point for the current state of the art in terms of data pre-processing for land surface applications. After accepting the premise of the paper, I had very few comments regarding its content. The toolkit is very well presented. I am therefore suggesting only very minor comments and clarifications and believe the article is a good fit for GMD.

Line 31: lots of acronyms here makes it quite hard to read. Could you maybe write out MDF its only used 6 times? Section 2 Background. A few additional examples of data processing environments designed to support large scale modelling would be a nice addition. Is it really the case that you would only classify the WRF as relevant to this broad definition? For me this section doesn't do enough to set the context within which LDT has been developed and is my only substantive criticism of the paper. Furthermore, you could also replace ArcGIS and Matlab with QGIS and R to have much the same functionality in an open source framework. Overall this paragraph is not very convincing relative to the rest of the paper. Is the text wonky on Figure 1? Maybe it's just my eyes, but it would look a little better if straightened up. P6 line 28: Am I correct in thinking that hydrological response unit approach to sub-grid parametrisation is not supported. If so could you briefly comment on the implications and future potential/challenges in this regard? Section 4.4. The frequent use of currently here suggest changes are planned or in progress. Perhaps either mention imminent development plans or drop the "currently" bit.

---

## Author Comment (AC3) · 7 Jul 2018

We would like to thank the reviewer for their thorough review of the paper, comments and helpful feedback.

Comment #1: Line 31: lots of acronyms here makes it quite hard to read. Could you maybe write out MDF its only used 6 times?

Response: Without the page specified, we are not sure which "Line 31" the reviewer may be referring to.

In response to the reviewer's request about the "MDF" acronym, we do introduce what

the "MDF" stands for on page 3, line 6, but we have included it again on line 25 of page 4 and in our Figure 1 caption.

Comment #2: Section 2 Background. A few additional examples of data processing environments designed to support large scale modelling would be a nice addition. Is it really the case that you would only classify the WRF as relevant to this broad definition? For me this section doesn't do enough to set the context within which LDT has been developed and is my only substantive criticism of the paper. Furthermore, you could also replace ArcGIS and Matlab with QGIS and R to have much the same functionality in an open source framework. Overall this paragraph is not very convincing relative to the rest of the paper.

Response: Thank you for the feedback on this particular section. In our review of available data processing software for land surface and hydrological models, not many models have a designated and comprehensive preprocessor that handles all inclusively many of the features that LDT does or have supporting documentation available. The WRF preprocessing toolset is one known example that has technical description documents, tutorials and available test cases. Also, we are aware of some LSM or hydrological model preprocessing software developed by different institutions, but in some instances, some of the documentation may not be shared publicly (e.g., with the JULES model) or several different steps, scripts and languages may be required to derive new inputs for a model (e.g., the Community Land Model).

We do agree that providing a few additional examples of data processing environments would be useful to further highlight the purpose for why LDT was developed. We have updated the first paragraph of Section 2, addressing the concerns noted by the reviewer, and included some additional examples of what is found with other pre-processing environments.

In response to the reviewer's point about QGIS and R being an open-source alternative to ArcGIS and Matlab, this is a valid point, so we have modified the text in this part of

Section 2 to better reflect these other options.

Comment #3: Is the text wonky on Figure 1? Maybe it's just my eyes, but it would look a little better if straightened up.

Response: Thank you for noticing this. We will make sure that the Figure 1 graphic is updated for the final paper submission.

Comment #4: P6, line 28: Am I correct in thinking that hydrological response unit approach to sub-grid parametrisation is not supported. If so could you briefly comment on the implications and future potential/challenges in this regard?

Response: We appreciate the reviewer's question on this topic. At this time, we do not support the use or models that use hydrological response units (HRUs) in any public versions of LDT. LDT does provide support for gridded drainage basin areas for the HyMAP-1 and 2 model versions. However, efforts are underway to merge WRF-Hydro with LDT and the Land Information Systems (LIS), where WRF-Hydro utilizes Hydrologic Unit Codes (HUC) basin and stream segments, so this could lead to future development and incorporation of models that support the HRU model unit.

The potential may exist for representing HRUs and using LDT to support that. Currently, ArcGIS is used to help derive some of the needed topographic parameters and basin delineation to generate the higher resolution routing grids (e.g., for WRF-Hydro). Thus, the challenge will be finding ways to replicate the ArcGIS capabilities within the LDT software environment to be able generate the relevant subbasin-related information. Also, Samaniego et al. (2017) point out some of efforts and difficulties associated with employing the HRU as the landunit representation involve issues, for example, with scaling of parameters to different domains or coarser resolutions not calibrated. So as the need arises, we would make the effort to implement the necessary support for HRU subgrid parameterization as best as possible, trying to address some of these challenges.

Comment #5: Section 4.4. The frequent use of "currently" here suggest changes are planned or in progress. Perhaps either mention imminent development plans or drop the "currently" bit.

Response: Thank you for this suggestion. We have replaced "currently" in a couple of locations with different wording to indicate whether a feature is in progress or fully mature at this time.

References

Samaniego, L., and co-authors: Toward seamless hydrologic predictions across spatial scales, Hydrol. Earth Syst. Sci., 21, 4323-4346, 2017. https://doi.org/10.5194/hess-21-4323-2017

---

## Author Response (AR1)

**Response to Referees**

Title: The Land surface Data Toolkit (LDT v7.2) - a data fusion environment for land data assimilation systems
Author(s): Kristi R. Arsenault et al.
MS No.: gmd-2018-63
MS Type: Model description paper

**A) Referee #1 Comments:**
"The authors have described a specialized land model processing tool, the NASA Land Information System Data Toolkit (LDT). This tool has been developed to support applications of the Land Information System, so it is more usefully viewed as a modeling system specific pre-processor than as a software tool for general use by the hydrological community. Nevertheless, LDT should be of interest to journal readers both because LIS and its supported Land Surface Models are widely used by the community and because the classes of processing capabilities included in LDT are of general interest for many advanced hydrological modeling systems. The manuscript appears to provide an accurate description of LDT, and the level of C1 detail is adequate for the reader to understand the motivation and capabilities of the tool. I am happy to recommend the manuscript for final publication without further modification."

**Response to Referee #1:**
We thank the referee for their thorough review of our manuscript and for the positive feedback. We just want to quickly note that we do develop LDT not only to support the Land Information Systems (LIS) software but also the broader land modeling and hydrology community.

**B) Referee #2 Comments:**
"This paper presents an overview of the land surface data toolkit v7.2. The main purpose of the paper is to explain the functionality of the toolkit and present the justification for various components with reference to the literature. Such tools are essential to the community as a means of facilitating model development and implementation, especially given the ever-increasing availability of observational data and computing resources. The paper is not a technical description of the toolkit as such, more a very detailed description of the components. I was a little sceptical about the value of the paper for this reason, however after reading the content I believe it to be a worthy contribution C1 to the literature and reference point for the current state of the art in terms of data pre-processing for land surface applications. After accepting the premise of the paper, I had very few comments regarding its content. The toolkit is very well presented. I am therefore suggesting only very minor comments and clarifications and believe the article is a good fit for GMD."

**Response to Referee #2:**

We would like to thank the referee for their thorough review of the paper, comments and helpful feedback.  Please find below our responses to specific comments and questions made by the referee.

**R2: #1)  Line 31: lots of acronyms here makes it quite hard to read. Could you maybe write out MDF its only used 6 times?**

Without the page specified, we are not sure which "Line 31" the reviewer may be referring to.

In response to the reviewer's request about the "MDF" acronym, we do introduce what the "MDF" stands for on page 3, line 6, but we have included it again on line 31 of page 4 and in our Figure 1 caption.

**R2: #2)  Section 2 Background. A few additional examples of data processing environments designed to support large scale modelling would be a nice addition. Is it really the case that you would only classify the WRF as relevant to this broad definition? For me this section doesn't do enough to set the context within which LDT has been developed and is my only substantive criticism of the paper. Furthermore, you could also replace ArcGIS and Matlab with QGIS and R to have much the same functionality in an open source framework. Overall this paragraph is not very convincing relative to the rest of the paper.**

Thank you for the feedback on this particular section.  In our review of available data processing software for land surface and hydrological models, not many models have a designated and comprehensive preprocessor that handles all inclusively many of the features that LDT does or have supporting documentation available.  The WRF preprocessing toolset is one known example that has technical description documents, tutorials and available test cases.  Also, we are aware of some LSM or hydrological model preprocessing software developed by different institutions, but in some instances, some of the documentation may not be shared publicly (e.g., with the JULES model) or several different steps, scripts and languages may be required to derive new inputs for a model (e.g., the Community Land Model).

We do agree that providing a few additional examples of data processing environments would be useful to further highlight the purpose for why LDT was developed.  We have updated the first paragraph of Section 2 with the following text (which is also highlighted in the revised manuscript):

"One example includes the Community Land Model, versions 4 and higher (Oleson et al., 2010), which has data preprocessing scripts and online instructions provided to users to generate inputs for the model.  The developers provide standardized global input files, but if the user wants to run for another resolution, regional subset or use different parameters (e.g., landcover map), the user must modify and run several different scripts to generate the necessary input files, which can take several steps. Other examples include the National Center for Atmospheric Research (NCAR) WRF Preprocessing System (WPS) and the pre-processor for the WRF Hydrological modelling extension (WRF-Hydro; Gochis et al., 2014; Sampson and Gochis, 2015). WPS offers a suite of specific datasets and primarily serves the preprocessing needs of the WRF community (Skamarock et al, 2008) and some in the Noah land surface model community (e.g., Chen et al., 2007). If the user wants to use WPS for Noah model parameter preprocessing, the user is either limited to what preprocessed parameters are available, or they have to generate those files to be in the specific WPS required format before using them."

In response to the reviewer's point about QGIS and R being an open-source alternative to ArcGIS and Matlab, this is a valid point. We decided to remove the statement in the revised manuscript, since it did not flow as well with this paragraph, so we have modified the text in this part of Section 2 to the following:

"The WRF-Hydro preprocessor can utilize different hydrological-based topographical datasets, such as HydroSHEDS (Lehner et al., 2008), however the input elevation maps to WRF-Hydro preprocessor are expected to be specifically in ArcGIS raster format, a proprietary format (ESRI, 2016), and may require more testing and effort when using from open-source alternatives, like QGIS (http://qgis.osgeo.org)."

**R2: #3) Is the text wonky on Figure 1? Maybe it's just my eyes, but it would look a little better if straightened up.**

Thank you for noticing this. We will make sure that the Figure 1 graphic is updated for the final paper submission.

**R2: #4)  Page 6, line 28: Am I correct in thinking that hydrological response unit approach to sub-grid parametrisation is not supported. If so could you briefly comment on the implications and future potential/challenges in this regard?**

We appreciate the reviewer's question on this topic. At this time, we do not support the use of hydrological response units (HRUs) in any public versions of LDT, but there are plans to support HRUs and also in general support for unstructured grids. There are certain land surface models (e.g. NCAR Structure for Unifying Multiple Modeling Alternatives; SUMMA) that are specifically designed to utilize notions of HRUs. Efforts are currently underway to include SUMMA within LIS. To support this implementation, LDT will be enhanced with the ability to represent notions of HRUs.

Samaniego et al. (2017) point out some of efforts and difficulties associated with employing the HRU as the landunit representation involve issues, for example, with scaling of parameters to different domains or coarser resolutions not calibrated. Functionally, the key underlying change that would be needed is in the grid/map projection support. We will rely on community developed tools such as ESMF, to enable support for the required grid transformations for a large suite of grids.

No modifications to the manuscript text were made in relation to our response.

**R2: #5) Section 4.4. The frequent use of "currently" here suggest changes are planned or in progress. Perhaps either mention imminent development plans or drop the "currently" bit.**

Thank you for this suggestion. We removed the first "Currently" on page 10, line 30, and replaced the second "Currently" on page 11, line 1, with "At this time".

**References**

Samaniego, L., *and co-authors*: Toward seamless hydrologic predictions across spatial scales, *Hydrol. Earth Syst. Sci.,* 21, 4323-4346*, 2017.* https://doi.org/10.5194/hess-21-4323-2017

[revised manuscript text omitted]